# Italian Canyoning Guides: Physiological Profile and Cardiometabolic Demand during Rope Activities

**DOI:** 10.3390/sports12050129

**Published:** 2024-05-13

**Authors:** Tommaso Di Libero, Lavinia Falese, Stefano Corrado, Beatrice Tosti, Pierluigi Diotaiuti, Angelo Rodio

**Affiliations:** Department of Human, Social and Health Sciences, University of Cassino and Southern Lazio, Campus Folcara, Via S. Angelo, 03043 Cassino, FR, Italy; l.falese@unicas.it (L.F.); stefano.corrado@unicas.it (S.C.); beatrice.tosti@unicas.it (B.T.); p.diotaiuti@unicas.it (P.D.); a.rodio@unicas.it (A.R.)

**Keywords:** canyoning, fitness level, motor abilities, safety, physiological profile, support simulation

## Abstract

Canyoning activities require physical effort, highlighting the importance of maintaining a proper physical fitness. Canyoning guides emerge as key figures, not only to ensure safety during the experience but also to handle unforeseen situations promptly. This study aims to assess the physiological profile of canyoning guides and the cardiorespiratory demands experienced during rope activities by means of oxygen uptake and heart rate measurements. Seventeen canyoning guides (42.6 y ± 10.78; BMI of 24.0 kg/m^2^ ± 2.95) carried out coordinative and conditional tests. The participants showed good values in strength tests (27.3 cm ± 5.97 and 23.3 rep ± 8.06 in SJ and PUp tests, respectively), while the flexibility of males and females was below and well above the average, respectively. A noteworthy result was observed in the reaction test, in which a better performance was recorded with the non-dominant hand (168.1 ms vs. 202.0 ms). All subjects exhibited a low aerobic capacity by means of an RD test (10.6 ua ± 6.62). During rope activities and emergency/support simulations, metabolic and cardiovascular data indicated that a moderate/high effort was exerted, confirmed by an oxidative stress analysis. In conclusion, this study demonstrated how canyoning guides face significant physical requirements, but their physiological profile regarding aerobic power was not appropriate. Therefore, these findings could offer valuable insights into the development of specific training to ensure an appropriate aerobic fitness to perform canyoning safely.

## 1. Introduction

In recent years, outdoor sports, particularly wild sports, have become increasingly popular as they offer an opportunity for recreation and lead to numerous benefits, including stress reductions and an improved physical and mental well-being [1,2,3]. In addition, they are also carried out in pristine environments far from urban areas; performing these activities in the presence of airborne particulate matter decreases performance and poses a potential risk of cardiovascular and pulmonary diseases [4]. Outdoor sports involve various activities, such as climbing and rappelling, contributing to the development of strength, endurance and muscular flexibility [5]. These outdoor activities, like many others indoors, improve coordinative and conditional abilities [6]. Moreover, they are suitable for practice individually, in the company of friends or as part of corporate groups to encourage the development of teamwork [7]. Canyoning represents an emerging movement that combines several activities typical of adventure sports, including hiking, swimming, diving, mountaineering, orienteering and rope techniques to navigate a challenging terrain. Moreover, the experience of navigating through gorges and facing natural obstacles contributes to improving self-esteem [8]. Although not a competitive sport, canyoning provides an engaging experience that stimulates the cardiorespiratory and muscular systems, facilitating a performance response [9,10]. Various indirect field tests can be employed to evaluate these aspects. In this scenario, the canyoning guide plays a pivotal role, guaranteeing safety throughout the experience and handling unforeseen situations with promptness and physical prowess [11]. In this context, the canyoning guide has a key role and is called upon to ensure safety during the experience or in unexpected situations requiring readiness and physical effort [12]. There are no studies on the overall energy expenditure of canyoning, aside from its traumatic effects. Canyoning involves hiking, swimming, diving, mountaineering, orienteering and rope techniques, resulting in varying levels of energy expenditure [13]. Canyoning is a physically demanding activity that engages major muscle groups, including the upper limbs, lower limbs, and core. The upper limbs are crucial for gripping and holding, in turn essential for climbing and rappelling, while the lower limbs provide stability and strength for jumping and moving across rugged terrain. Core muscles are vital for balance and stability on uneven and slippery surfaces in canyons [14]. The intensity and duration of canyoning vary, often lasting several hours, with activities from moderate walking to high-intensity climbing, jumping, and swimming [15]. Therefore, it becomes imperative for guides to conduct vigorous physical preparation. In fact, canyoning guides must be physically fit and capable of enduring prolonged cardiovascular and muscular stress [16]. Environmental factors like cold water, variable weather, and high altitudes add physiological stress, increasing metabolic rates and demanding rapid bodily adaptation to maintain core temperature and muscle efficiency [17]. Canyoning guides skilled in handling specific emergencies become true experts in essential techniques for dealing with the unpredictable risks associated with canyons, including sudden changes in weather and accidental falls related to the environment [18,19]. The guides’ physiological characteristics play a critical role, especially when traditional means of rescue may be unavailable in remote canyons [20]. This study explored guides’ main physiological parameters in the canyoning context, focusing on assessing physiological profiles through indirect tests of coordinative and conditional abilities. Cardiorespiratory parameters and metabolic effort were measured during simulated ascent and descent on a wall. This research aimed to evaluate the physical fitness of canyoning guides by analyzing their cardiorespiratory and metabolic capacity while undertaking a simulated canyoning task to determine their ability to adapt to the strenuous requirements of the sport, with a specific emphasis on their coordinative and conditional abilities [21]. The findings could be used as guidelines for training programs or as supplementary tests to technical abilities, emphasizing the importance of possessing a good technique and physical fitness. Enhancing the training of guides could significantly reduce the risk of accidents during canyoning, thus helping to increase the overall safety of the activity.

## 2. Materials and Methods

Through the National Canyoning Guides Organization (ENGC), which boasted 60 members, 17 subjects voluntarily participated. A diverse group of 17 guides, consisting of 13 men and 4 women with varying experience levels, were selected for this study. Anthropometric measurements regarding weight, height and anatomical circumferences were taken to assess body composition indirectly (Table 1). Instruments such as a statimeter (Tanita, TBF-300) for measuring height and a scale for weight were utilized. Each individual’s body mass index (BMI) was calculated using a specific equation as indicated by Nuttal et al. [22]. The body fat mass (FM) percentage was estimated by integrating body circumferences into a dedicated Equation [23]. Subsequently, a battery of indirect tests were conducted to evaluate the physical fitness of the canyoning guides. These tests assessed their dexterity, balance, strength, and endurance, all essential canyoning abilities. After completing the tests, some participants engaged in a simulated canyoning activity that involved wall climbing. This exercise provided a realistic context for evaluating physical performance in situations typical of canyoning, especially in support or emergency scenarios. Many guides prioritize technical skills, but there is a growing recognition that assessing their physiological characteristics is equally important to ensure they can meet the demands of canyoning. Coordinative and conditional abilities were evaluated through a battery of tests that included several assessments, taking into account the corresponding reference tables divided by gender and age. The battery tests included the V-Sit and Reach (VSR) [24] test for flexibility, ranging from 12 (low) to 30 (very good); the Stork Balance Stand (SBS) test [25] for both the dominant (SBSDL) and non-dominant leg (SBSNDL) to assess balance; the ruler test (RT) [26] for hand–eye coordination and reaction times, performed for both dominant (RTDH) and non-dominant hands (RTNDH); the Squat Jump (SJ) test [27] for lower limb strength; the Push-Up (PUp) test [28] for upper limb strength; and the Ruffier–Dickson (RD) test [29] for the cardiovascular system. In the VSR test, participants sat on the floor with their legs stretched forward and slowly bent forward as far as possible, with the distance achieved being recorded on a centimeter strip. In the SBS test, participants stood on one foot with the free foot resting on the inner knee of the standing leg and arms placed on the hips, and balance was maintained for as long as possible, with time recorded using a stopwatch. In the RT test, a ruler was held vertically, suspended between the outstretched fingers of a participant. The ruler was then dropped without warning, and the participant must catch it between their fingers as quickly as possible. The distance the ruler travels before being caught is recorded and converted into meters per second. In the SJ test, participants performed a vertical jump starting from a squatting position, with the height of the jump quantified using the My Jump 2 app (My Jump 2, Apple Inc., Cupertino, CA, USA). In the PUp test, participants performed as many push-ups as possible until their muscles were fatigued, with the total number of successful push-ups being recorded. Finally, in the RD test, an index was calculated to assess the efficiency of the participant’s cardiovascular recovery. This test measures the cardiovascular recovery capacity after exercise. It begins by taking the subject’s resting heart rate, measured in beats per 10 s and multiplied by 6 to obtain bpm, denoted as *F*. After performing 30 leg bends in 45 s, the subject sits down and the heart rate is measured immediately, F1, and again after one minute, F2. The test value *T* is calculated with the formula $T=\frac{\left(F1-70)+2\times (F2-F\right)}{10}$, which assesses how quickly the heart returns close to its resting frequency. This test provides useful insight into cardiovascular health and the heart’s response to exertion. Heart rate was calculated using the Polar H10 (Polar Electro Oy, Kempele, Finland). All tests were performed sequentially thrice, respecting rest periods as required in the protocol. Subsequently, the RD and PUp tests were administered once each. After assessing their motor abilities, six experienced subjects were selected to perform a cardiorespiratory and metabolic test that involved simulating a rope activity on a 25-m vertical wall. The rope activity consisted of (1) wearing technical equipment, (2) hiking to the top of the wall, and (3) three descents and two alternating ascents. Cardiorespiratory and metabolic parameters were measured using a metabolimeter (COSMED K5, B^2^_,_ Pavona, Roma, Italy) during the rope activity. The protocol included two main activities, a rope ascent and descent, structured to simulate a possible real-world scenario related to the sport. The speed of the ascent was adjusted individually to suit each participant’s physical abilities and comfort level. Each ascent lasted from 5 to 10 min, depending on the participant’s pace, and each descent, on the other hand, did not exceed 30 s, as participants used their body weight and the force of gravity to facilitate a fast and smooth descent. Moreover, urine sampling was performed to assess oxidative stress using parameters like free radical concentration, specifically Reactive Oxygen Species (ROS) and Glutathione Disulfide (GSSG). This assessment offered insights into the temporal dynamics of oxidative stress in response to physical activities of different durations post exertion [30]. Urine sample collection was carried out in the morning before breakfast, referred to as baseline (T0); before putting on the equipment and performing the rope exercise, referred to as pretest (T1); and 90 min and 3 and 5 h after the end of the rope exercise, referred to as T2, T3 and T4, respectively. Further analysis was conducted to assess oxidative stress through ELISA tests [31]. A total of 10 mL of urine was collected from six subjects enrolled in the study. Samples were collected immediately upon awakening and on an empty stomach (T0), just before the start of the activity (T1), and 1.30 h (T2), 3 h (T3), and 4 h after the end of the activity (T4). The amount of Oxidized Glutathione (GSS); total Glutathione (GSS + GSH); 8-hydroxy deoxyguanosine (8-OHdG), a marker of DNA damage; and creatinine in the collected urine was determined using enzyme-linked immunoadsorbent assay kits (Thermo Fisher Scientific, Waltham, MA, USA) according to the manufacturer’s instructions. Optical density (OD) was measured at 450 nm using a microplate reader (NB-12-0035, NeBiotech, Holden, MA, USA). The amount of GSS, GSS + GSH and 8-OHdG is expressed as the creatinine ratio.

## 3. Results

All parameters were expressed as means, relative SD, and *p*-values, as shown in Table 1 and Table 2. The Shapiro–Wilk test for data normalization was performed. In addition, a comparison between the dominant and non-dominant limbs for SBS and RT tests was also conducted using the *t*-test to assess significant differences in the measurements obtained between the two limbs. Statistical significance was set to p≤0.05. Out of all the participants, the majority (about 65%) had over a decade of experience in this field. According to the BMI results, over 76.47% of participants had normal weight values. Additionally, the FM% showed that males had a good value (18.4% ± 5.6), while females had a very lean value (14.0% ± 3.9). Regarding coordinative and conditional abilities, the VSR results were below average (19.6 cm ± 10.2) and well above average (27.1 cm ± 11.73) for males and females, respectively. Both genders presented low balance values in the SBS test, with the dominant leg (5.2 s ± 2.97) higher than the non-dominant one (3.6 s ± 1.8). Good and excellent results were exhibited in the RT test with dominant (202.0 ms ± 45.60) and non-dominant hands (168.1 ms ± 34.09). For the SJ test, males showed an excellent value (27.3 cm ± 5.97), and females had a very good value (22.0 cm ± 4.57). PUp results indicated very good and average levels for males (23.3 rep ± 8.06) and females (18.2 rep ± 4.8). At last, the RD results showed low adaptation (10.6 ua ± 6.62) in the aerobic fitness level upon measuring F, F1 and F2 (68.3 ± 15.26, 129.6 ± 26.16, and 91.5 ± 23.96, respectively). The oxygen consumption and heart rate trends measured during an emergency/support simulation indicate a significant increase in metabolic demand (30 mL·kg${\;\;}^{-1}$·min${\;\;}^{-1}$), as shown in Figure 1. The intensity, as indicated by an increase of about 9 METs, classifies the activity as medium height; the heart rate reserve also showed values indicative of medium/high-intensity exercise (80%), as shown in Figure 2. Moreover, Figure 3 shows how GSSG concentrations, an indicator of oxidative stress, varied significantly in the recovery phase after the rope activity. GSSG increases until a peak value in the T3 phase and then decreases in the T4 phase, 3 and 5 h after the simulated scenario, respectively. Although the exact values of the markers are not specified, their increase in response to the activity is indicative of increased metabolic stress.

## 4. Discussion

Canyoning guides are a crucial reference point for practitioners of this sport, constantly adapting to various situations that may present risks due to environmental and weather-related factors. The primary purpose of this study was to examine canyoning guides to determine their physiological profile as being technical experts and determine their suitability in meeting the cardiovascular and metabolic demands of performing this sport.

The prevalence of normal weight and other anthropometric body composition characteristics is defined as an overall good health status [32,33]. The VSR test results indicate that male participants exhibited a below-average flexibility, while female participants demonstrated a well-above-average flexibility. A lack of adequate flexibility also has its drawbacks. The rigidity and limited mobility of the trunk and pelvis can lead to an inadequate performance in motor tasks, resulting in energy expenditure and an increased risk of injury [34]. This highlights the need for a balanced approach to flexibility training tailored to individual needs. Those with excessive flexibility should enhance muscle strength and stability to support joints effectively. Conversely, those with limited flexibility would benefit from targeted exercises to increase joint mobility, reducing the risk of injury and improving overall performance. On the other hand, the excessive flexibility shown in females, characterized by an advanced range of motion without the necessary muscular strength and stability, can lead to joint instability, making joints more prone to misalignments, sprains, and ligament or tendon injuries during physical exertion [35]. This condition underscores the delicate balance needed in flexibility training, where excessive and insufficient flexibility presents several challenges [36]. The strategy of integrating training programs focused on strengthening the muscles responsible for joint stability promotes a balanced physical state by ensuring muscle strength and stabilizing flexibility, minimizing injury risks, and boosting overall performance [37]. SBS results showed no significant difference between the genders, recording poor balance scores in both cases. This finding is unexpected for a profession that often requires good balance, indicating a potential area for improvement in training programs. It is critical to note that differences in the results may not generally indicate an imbalance problem. Instead, it could be attributed to the particular nature of the test that is more suitable for dexterity sports, i.e., rhythmic gymnastics or fencing [38]. This test was designed to assess static balance, which may not align with the dynamic actions required by canyoning. Alternatively, it may be more appropriate to consider less stressful tests with the eyes open and the sole fully supported. In the RT, the non-dominant hand performed better than the dominant hand. This asymmetry could result from specific technical practices or activities involving the non-dominant hand, such as canyoning. Detailed analysis of the score distribution shows significant variability among the participants, highlighting that some individuals excel significantly while others exhibit a more moderate performance. This predominance of better results with the non-dominant hand could be related to their work as canyoning guides. In situations that require high dexterity and strength to grasp ropes, rocks, or similar objects (such as climbing equipment or safety harnesses), there may be greater stiffness in the hand most frequently used, i.e., the dominant hand. This may explain the counter-intuitive results of the RT test, in which the non-dominant hand showed superior performance. This diversity could be further explored to better understand the factors contributing to individual differences in reaction times. Test results of both genders’ upper and lower limb strengths (SJ, PUp) were well above the average, indicating that disciplined practice or specific training approaches were appropriate for these conditional abilities. All subjects were above average regarding strength, and only a few fell below the median. It is possible to assert that this sport promotes good overall physical fitness, as observed in the canyoning guides. The results from the RD test indicated that over 70% of the participants displayed a low adaptability. This finding prompts investigation into adopting specialized strategies to improve cardiorespiratory endurance within training regimens. The outcomes of the emergency simulations and rope activity scenarios highlight that the participants exerted significant aerobic effort, as evidenced by the high levels of oxidative stress markers. During the rope activities, the oxygen consumption reached approximately 30 mL·kg${\;\;}^{-1}$·min${\;\;}^{-1}$, i.e., about 9 METs [39], indicating that the intensity of sustained exertion was at or above the anaerobic threshold in subjects with a low aerobic capacity. The level of risk inherent in certain activities can vary greatly, as evidenced by the results of our simulations, the physiological profiles and the relative fitness levels. It is crucial to understand these factors in order to ensure safety. Our findings demonstrate that this knowledge can be particularly valuable in designing effective training programs that take into account factors such as supportive environments and changing weather conditions that can increase metabolic and cardiorespiratory demands. Endurance is a key component of activities such as cycling, running, and swimming, which are essential for improving aerobic power, a critical factor in the type of activity being studied. Our research identified coordination, muscle stiffness, and aerobic power as the most significant areas needing improvement among study participants, highlighting the importance of incorporating targeted exercises into their training. This study has some limitations due to the small sample size and lack of data on peak oxygen consumption. These constraints were compounded by the difficulty of traveling to the laboratory, which hindered comprehensive and in-depth data collection. The main intent was to use inexpensive field tests to take measurements before embarking on the different routes that guides face in the conditions of the natural environment. Nonetheless, the findings indicate that the subjects experienced a considerable level of exertion [40]. This research underscores critical factors for future training and physical fitness in comparable scenarios, underscoring the necessity for further exploration to better understand these intricacies. Several intervention strategies could address this critical problem, such as incorporating specific aerobic activities like running, cycling or swimming into training programs. The synthesis of the results of this study clarifies the critical physiological challenges faced by canyoning guides during high-hazard scenarios. By measuring oxygen consumption under simulated hazardous conditions, we gained a deeper insight into the physical demands placed on guides, highlighting the need for training programs that reflect the intense requirements of their role. The correlation between exercise intensity and increased oxidative stress, as evidenced by increased GSSG levels, further accentuates the importance of comprehensive recovery methods and nutritional interventions to mitigate the impact of oxidative stress. Notably, this study highlights areas where guides show deficiencies in their coordinative and conditional abilities, recommending a targeted approach in training regimens. In conclusion, this research contributes to understanding the physiological underpinnings critical to the performance of canyoning guides. It underscores the imperative of developing specialized training and recovery protocols to ensure their safety, health, and effectiveness in coping with the demands of their demanding profession.

## 5. Conclusions

In this sport, it is essential to use specialized difficulty rating tables to assess the difficulty of routes through their vertical and aquatic characteristics. These tables are necessary for guides to understand and prepare for the terrain. However, the effectiveness of these ratings goes far beyond the tables. Regarding guides, our results show that not only technical aspects but also fitness aspects are important factors to take into account to ensure a better and safer approach to activities. Training programs for guides should focus on developing technical skills through various levels of difficulty and include a broader spectrum of assessments and exercises. Performing coordinative and conditional ability tests is useful to provide insight into the trials a guide may tackle that have similar energy demands to the natural paths encountered during canyoning. Moreover, the training program should focus on responding effectively to rescue or support situations, simulating medium/high-energy technical maneuvers linked to both unpredictable environmental conditions and sudden changes in weather. In fact, simulating these conditions indicated the metabolic work required, which was around or up to the anaerobic threshold, inducing increased lactate metabolic acidosis and relative muscle fatigue. Indeed, specific medium/high-intensity aerobic activities such as biking, running, swimming, and hiking are suggested to be included in training programs.

## Figures and Tables

**Figure 1 sports-12-00129-f001:**
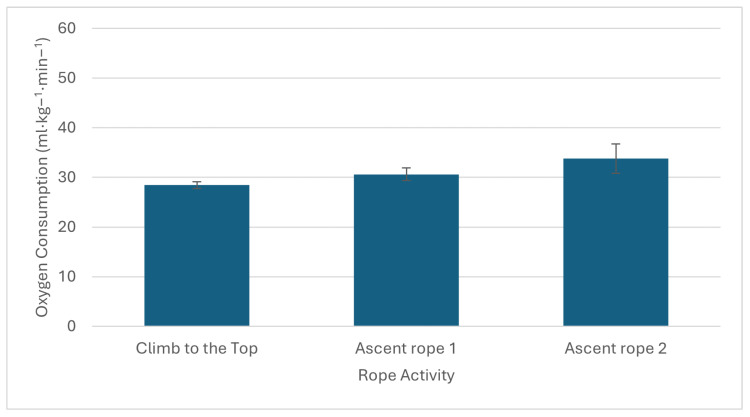
Oxygen consumption values measured during rope and emergency/support simulated activities. The oxygen consumption indicates a significant increase in metabolism of about 30 mL·kg${\;\;}^{-1}$·min${\;\;}^{-1}$ during the simulated condition: climb to the top and ascend rope 1 and 2. The peak oxygen uptake value reflects a high metabolic effort.

**Figure 2 sports-12-00129-f002:**
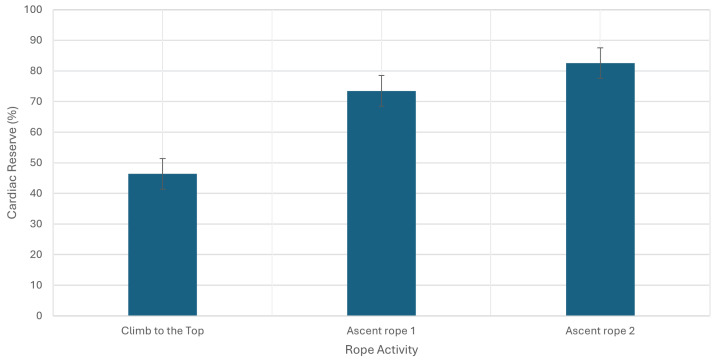
Heart rate reserve measured during emergency/support simulated activities. An indication of a moderate to high cardiovascular response can be observed when the values reach around 80%.

**Figure 3 sports-12-00129-f003:**
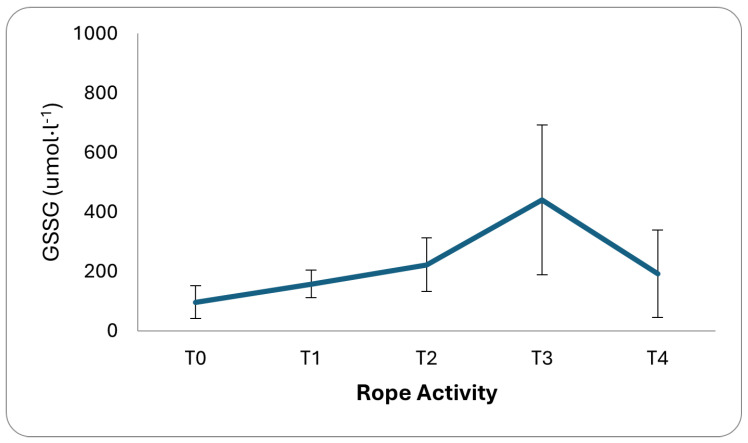
During the recovery phase after rope and emergency/support, changes in GSSG concentrations were observed. GSSG concentrations peaked during the T3 phase and then decreased in the T4 phase, corresponding to 3 and 5 h, respectively, after the simulated scenario.

**Table 1 sports-12-00129-t001:** Participant’s anagraphic and anthropometric values, expressed as mean and relative SD.

	Male	Female
Year experience (y)	8.1 ± 7.34	7.0 ± 3.92
Age (y)	43.4 ± 11.92	40.0 ± 6.38
Weight (kg)	79.5 ± 7.63	59.8 ± 7.41
Height (cm)	179.6 ± 5.36	168.0 ± 1.41
BMI (kg/m${\;\;}^{2}$)	24.9 ± 2.57	21.1 ± 2.34
FM (%)	18.4 ± 5.65	14.0 ± 1.92

**Table 2 sports-12-00129-t002:** Coordinative and conditional scores of the sample.

	Mean ± SD	Level	W	*p*
VSR (cm)	19.3 ± 13.04	Above average	0.978	0.936
SBSDL (s)	5.2 ± 2.97	Low	0.885	0.039
SBSNDL (s)	3.6 * ± 1.80	Low	0.853	0.012
RTDH (ms)	202 ± 45.60	Good	0.966	0.742
RTNDH (ms)	168.1 * ± 34.09	Excellent	0.947	0.415
SJ (cm)	27.3 ± 5.98	Very good	0.929	0.21
PUp (rep)	21.9 ± 6.98	Very good	0.968	0.79
RD Index (ua)	10.6 ± 6.63	Very low	0.967	0.771

* is indicative of statistical analysis differences between balance (SBS) and ruler test (RT) data obtained in dominant (SBSDL - RTDH) and non-dominant (SBSNDL - RTNDH) limbs, and the relative *p*-value was set to *p* < 0.05. Data are expressed as means, relative SD, Wilcoxon rank and *p*-value.

## Data Availability

No new data were created or analyzed in this study. Data sharing is not applicable to this article.

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
