# Peer review of "Italian Canyoning Guides: Physiological Profile and Cardiometabolic Demand during Rope Activities"

_sports, 2024, doi:10.3390/sports12050129_

Round 1

Reviewer 1 Report

Comments and Suggestions for Authors

Nice read about Canyoning and the interest in examining the physiological characteristics of such guides.

The content in Methods and Results are very lacking. The authors need to do a major revision on the following.

- Include the model and make of every instrument used for the measurements.

- The tests used are not common and therefore authors must describe them and not just label them and expect readers to refer to the associated references to know how they were conducted.

- The results need to be clearly presented in Table form. And Table & Figure information must be more comprehensive and self-sufficient. What does W in Table 2 refer to? Please elaborate on your comparison of results, i.e., how do you deem poor, average, good, etc. 

Comments on the Quality of English Language

Minor editing is needed.

Author Response

Thank you for taking the time to review our work and for providing valuable feedback. We have carefully considered your comments and made the necessary revisions to address the issues you raised.

Below in detail, point by point, the responses to your comments:

  • Include the model and make of every instrument used for the measurements.
  • Instrument name models have been added, particularly in lines 76, 105 and 116. Details about the use of the instruments have been added in the section Materials and Methods.
  • The tests used are not common and therefore authors must describe them and not just label them and expect readers to refer to the associated references to know how they were conducted.
  • In revising the manuscript, we have added a brief description for each test. Specifically, changes were made from line 89 to line 128.
  • The results need to be clearly presented in Table form. And Table & Figure information must be more comprehensive and self-sufficient. What does W in Table 2 refer to? Please elaborate on your comparison of results, i.e., how do you deem poor, average, good, etc. 
  • In order to improve the clarity of Table 2, we have updated the caption by specifying the parameters analyzed, particularly by explicitly specifying that the "W" stands for Wilcoxon rank-sum. The ratings (poor, average, good) were taken from each test's appropriate literature reference tables and divided by age and gender. In addition, the level ratings were added to Table 2.

Reviewer 2 Report

Comments and Suggestions for Authors

Dear Editor,

Thank you for the opportunity to review the manuscript "Italian Canyoning Guides: Physiological Profile and Cardiometabolic Demand During Rope Activities" submitted to the Sports journal. This study aims to assess the physiological profile of canyoning guides and the cardiorespiratory demands sustained during rope activities.

Overall, this is a well-designed and executed study that provides valuable insights into the physical requirements and demands faced by canyoning guides. The authors have thoroughly examined the coordinative, conditional, and cardiometabolic aspects of the canyoning guides' performance, using a comprehensive battery of tests.

General comments:

1.    The introduction provides a good background on the importance of outdoor sports and the role of canyoning guides. However, it could be further strengthened by discussing the specific physical and physiological demands of canyoning in more detail, providing a stronger rationale for the study.

2.    The methodology section is well-described but could benefit from more information on the justification for the chosen test protocols and their reliability/validity in the canyoning context.

3.    The results section is clear and comprehensive, with appropriate statistical analysis. However, the discussion could be expanded to provide a more in-depth interpretation of the findings and their implications for canyoning guide training and safety.

4.    The conclusions section is concise and highlights the key takeaways, but could be further developed to provide more specific recommendations for the development of training programs and assessment of canyoning guides.

Specific comments:

Abstract

1.    Consider providing more details on the specific physiological and metabolic parameters measured during the simulated rope activities, as these seem to be key findings of the study.

2.    Clearly state the main conclusions and recommendations for canyoning guide training and safety based on the study's findings.

Introduction

1.    Expand the discussion on the specific physical and physiological demands of canyoning, such as the muscle groups involved, the intensity and duration of activities, and the environmental factors that contribute to the physiological stress.

2.    Provide more context on the role and importance of canyoning guides in ensuring the safety and well-being of participants, and how their physiological profile may impact their ability to perform this role effectively.

Methods

1.    Justify the choice of the specific test protocols used to assess the coordinative, conditional, and cardiometabolic abilities of the canyoning guides, and provide information on their reliability and validity in the canyoning context.

2.    Describe in more detail the simulated rope activity protocol, including the duration, intensity, and specific tasks performed by the participants.

Results

1.    Expand the presentation of the results to include more detailed data and statistics, particularly for the simulated rope activity, such as the peak oxygen consumption values, heart rate responses, and changes in oxidative stress markers.

2.    Discuss any notable differences in the physiological responses between the different rope activity tasks (e.g., ascent vs. descent) or between the participants with different levels of experience.

Discussion

1.    Provide a more in-depth interpretation of the findings, particularly how the physiological profile of the canyoning guides may impact their ability to handle emergencies and support other participants during canyoning activities.

2.    Discuss the implications of the study's findings for the development of training programs and assessment protocols for canyoning guides, addressing both the technical and physiological aspects of their role.

3.    Acknowledge the limitations of the study, such as the small sample size and the use of simulated activities, and how these may have influenced the findings.

Conclusions

1.    Provide more specific and actionable recommendations for the development of training programs and assessment protocols for canyoning guides, based on the study's findings.

2.    Emphasize the importance of incorporating both technical and physiological assessments in the selection and training of canyoning guides to ensure their safety and effectiveness in the field.

Overall, this is a well-designed and informative study that contributes to our understanding of the physical and physiological demands faced by canyoning guides. With the suggested improvements, this manuscript would be a valuable addition to the literature on outdoor sports and guide training.

Sincerely,

Author Response

Thank you very much for your thoughtful and constructive review. We really appreciate your insights and suggestions on the manuscript.

General comments:

  1. The introduction provides a good background on the importance of outdoor sports and the role of canyoning guides. However, it could be further strengthened by discussing the specific physical and physiological demands of canyoning in more detail, providing a stronger rationale for the study.

  • Modifications have been made to the introduction. Canyoning's specific physical and physiological demands are now discussed in detail. These changes have been elaborated upon in the 'Specific Comments'

  1. The methodology section is well-described but could benefit from more information on the justification for the chosen test protocols and their reliability/validity in the canyoning context.

Changes have been implemented in the methodology section. Additional information on the justification for the chosen test protocols has been included. For more details, please see the 'Specific Comments' section.

  1. The results section is clear and comprehensive, with appropriate statistical analysis. However, the discussion could be expanded to provide a more in-depth interpretation of the findings and their implications for canyoning guide training and safety.

  • The results section has been revised and updated. The discussion has been expanded to provide a more in-depth interpretation. These additions are detailed in the 'Specific Comments'

  1. The conclusions section is concise and highlights the key takeaways, but could be further developed to provide more specific recommendations for the development of training programs and assessment of canyoning guides.

  • Enhancements have been made to the conclusions section and can be found in the 'Specific Comments'

Specific comments:

Abstract

We appreciate your suggestions and have carefully considered them within the constraints of the word limit for the abstract. Accordingly, we have made the following revisions:

  1. Consider providing more details on the specific physiological and metabolic parameters measured during the simulated rope activities, as these seem to be key findings of the study.

  • In line 3 of the abstract, we have now specified the main parameters that define the physiological profile, providing clarity on the focus of our research.

  1. Clearly state the main conclusions and recommendations for canyoning guide training and safety based on the study's findings.

  • In line 14, we have updated the text to reflect a more precise interpretation of our findings. The original sentence, "Therefore, it is critical to provide specific training to ensure aerobic fitness to perform canyoning safely," has been revised to "Therefore, findings could offer valuable insights into the development of specific training to ensure aerobic fitness to perform canyoning safely."

Introduction

  1. Expand the discussion on the specific physical and physiological demands of canyoning, such as the muscle groups involved, the intensity and duration of activities, and the environmental factors that contribute to the physiological stress.

  • We agreed with what was suggested and expanded the discussion to include the specific physical and physiological demands of canyoning. Line 40 to line 48: We updated the introduction, providing a complete description of canyoning's physical and physiological requirements. Our revisions include the muscle groups used, the duration and intensity of activities, and environmental factors that may affect physical stress.

  1. Provide more context on the role and importance of canyoning guides in ensuring the safety and well-being of participants, and how their physiological profile may impact their ability to perform this role effectively.

  • Line 48 to line 53: We have also expanded on the importance of canyoning guides and their crucial role in ensuring the safety and well-being of all participants.

Methods

  1. Justify the choice of the specific test protocols used to assess the coordinative, conditional, and cardiometabolic abilities of the canyoning guides, and provide information on their reliability and validity in the canyoning context.

  • To determine the fitness level of canyoning guides before starting a trail, we selected practical, specific and cost-effective tests. These validated tests assess coordinative, conditional, and cardiometabolic capabilities and can be administered in the field. Our goal was to ensure that guides were well prepared for the demands of their chosen trail while keeping the testing procedures accessible and efficient.

  1. Describe in more detail the simulated rope activity protocol, including the duration, intensity, and specific tasks performed by the participants.

  • We have integrated the required information into the text from line 123 to line 129.

Results

  1. Expand the presentation of the results to include more detailed data and statistics, particularly for the simulated rope activity, such as the peak oxygen consumption values, heart rate responses, and changes in oxidative stress markers.

  • Line 118: In the materials and methods section, we illustrated the specific type of intent required for rope ascending and descending activities.

  1. Discuss any notable differences in the physiological responses between the different rope activity tasks (e.g., ascent vs. descent) or between the participants with different levels of experience.

  • Lines 164 and 169: These values are not differentiated by experience level but indicate average intense effort.

Discussion

  1. Provide a more in-depth interpretation of the findings, particularly how the physiological profile of the canyoning guides may impact their ability to handle emergencies and support other participants during canyoning activities.

  • Line 232 to line 237: We implemented the information in the discussion part according to your helpful suggestions.

  1. Discuss the implications of the study's findings for the development of training programs and assessment protocols for canyoning guides, addressing both the technical and physiological aspects of their role.

  • Line 237 to line 240: Following your constructive advice, we updated the discussion section accordingly.

  1. Acknowledge the limitations of the study, such as the small sample size and the use of simulated activities, and how these may have influenced the findings.

  • Line 240 to line 244: We have integrated your useful recommendations into the discussion part.

Conclusions

  1. Provide more specific and actionable recommendations for the development of training programs and assessment protocols for canyoning guides, based on the study's findings.

  • Line 268 to line 272: In response to your requests, we refined the conclusion section to clarify the objectives of our study better.

  1. Emphasize the importance of incorporating both technical and physiological assessments in the selection and training of canyoning guides to ensure their safety and effectiveness in the field.

  • Line 272 to line 281: We have integrated the conclusion section to clarify the objectives of this work according to your requests.

Round 2

Reviewer 1 Report

Comments and Suggestions for Authors

Thank you for your revisions